# Trust Prophet or Not? Taking a Further Verification Step toward Accurate Scene Text Recognition

## ABSTRACT

Inducing linguistic knowledge for scene text recognition (STR) is a new trend that could provide semantics for performance boost. However, most autoregressive STR models optimize one-step ahead prediction (i.e., 1-gram prediction) for character sequence, which only utilizes the previous semantic context. Most non-autoregressive models only apply linguistic knowledge individually on the output sequence to refine the results in parallel, which do not fully utilize the visual clues concurrently. In this paper, we propose a novel language-based STR model, called ProphetSTR. It adopts an n-stream attention mechanism in the decoder to simultaneously predict the next $n$ characters based on the previous predictions at each time step. It behaves like a prophet, encouraging the model to predict more accurate results by utilizing the previous semantic information and the near future clues. If the prediction results for the same character at successive time steps are inconsistent, we should not trust any of them. Otherwise, they are reliable predictions. Therefore, we propose a multi-modality verification module, masking the unreliable semantic features and inputting with visual and trusted semantic ones simultaneously for masked prediction recovery in parallel. It learns to align different modalities implicitly and considers both visual context and linguistic knowledge, which could generate more reliable results. Furthermore, we propose a multi-scale weight-sharing encoder for multi-granularity image representation. Extensive experiments demonstrate that Prophet-STR achieves state-of-the-art performances on many benchmarks. Further ablative studies prove the effectiveness of our proposed components.

## CCS CONCEPTS

• **Computing methodologies** → *Scene understanding*; • **Applied computing** → **Document analysis**; **Optical character recognition**.

## KEYWORDS

Scene text recognition, Multi-modality verification, Language model, Multi-scale

## 1 INTRODUCTION

Scene text recognition (STR) task involves visual and semantic modalities, so it is a critical research topic bridging computer vision

*ACM MM, 2024, Melbourne, Australia*
© 2024 Copyright held by the owner/author(s). Publication rights licensed to ACM.
ACM ISBN 978-x-xxxx-xxxx-x/YY/MM
https://doi.org/10.1145/nnnnnnn.nnnnnnn

and language model (LM). It has been widely used for many applications, e.g., document analysis [36], cross-modal retrieval [35], autonomous driving [9], and text-based attacks [40].

Recently, many deep learning-based STR methods have been proposed. Most of them follow the paradigm of the encoder-decoder pipeline, where the encoder is responsible for visual feature embedding and the decoder for character sequence prediction. As the text owns linguistic characteristics, the linguistic information is generally used in STR models to improve recognition performance, especially for challenging cases such as occlusion, blur, and distortion.

Typically, there are two streams to use the linguistic information for STR. One stream is the auto-regressive (AR) STR [4, 23] generally adopting the generative LM [2]. It estimates the probability distribution of the next character based on the previous predictions at each time step. It is widely used for sequence modeling and sequence-to-sequence (Seq2Seq) learning [25, 31]. The second steam is the non-auto-regressive (NAR) STR [8, 24] that applies visually independent LM to refine recognition results from the vision model. It could output the prediction sequences in parallel by considering the bidirectional context.

However, the AR STR methods learn limited linguistic knowledge since they only achieve context from one direction. An intuitive way to capture the previous and future information is to construct a bidirectional model [21, 32], which merges the results from a left-to-right decoder and a right-to-left decoder. However, since the bidirectional decoders model the semantic features in different directions respectively, they could not share the same weights by using a single decoder. The ensemble models mean the twice amount of parameters.

To tackle this problem, we propose a Transformer-based model, called ProphetSTR, to effectively model the linguistic rules of scene text, leveraging both previous semantic context and future clues with a single prophet decoder (PD). It predicts future n-gram (n>1) characters simultaneously at each time step. In addition to the traditional LM or Seq2Seq model in STR approaches that optimize one-step-ahead character prediction, the prophet decoder learns n-step-ahead prediction. This future n-gram prediction serves as extra guidance that explicitly encourages the model to plan for future tokens and make decisions based on both previous context and future clues.

The architecture of the PD is based on the original Transformer-based decoder [33], which is composed of masked multi-head self-attention blocks, cross-attention blocks, and feed-forward layers. Instead, we introduce $n$ learnable tokens interacting with previous prediction information for future n-gram prediction. The weights are shared for each stream in the decoder. Thus, there is no great increase in the model size.

Each character in the scene text image will be predicted $n$ times by referring to different context information. The prediction results

may be unreliable if they are inconsistent. Otherwise, they can be trusted. Inspired by the training strategy, i.e., Masked Language-aware Module (MLM), in Natural Language Processing (NLP) methods [5, 43], we propose to use an additional multi-modal-based module to further verify the prediction results based on the trusted outputs and vision features. The unreliable prediction results are masked and input to the verification module with trusted character embeddings and position-aware visual tokens. The objective of the module is to predict the true character category of the masked tokens based on the vision clues and LM simultaneously. Since the trusted characters provide partial order-aware semantics of the text, the position-aware visual features could be aligned with them implicitly. It benefits the module to learn visual-related LM. Unlike most NAR STR methods that implement iterative steps [8, 23], this verification module is only executed a single time, which is sufficient to correct the prediction errors.

Furthermore, we explore more rational visual representations of scene text images and propose a Transformer-based multi-scale encoder to further boost the STR performance. Most visual feature encoders follow the pipeline of Vision Transformer (ViT) [6] by splitting each image into a sequence of pre-defined size fixed patches and then inputting the patches to Transformer layers after the patch embedding process. The experimental results demonstrate the patch size effect on the classification accuracy of different datasets. Therefore, instead of using a fixed patch resolution, we use multi-scale patch representations, enabling the Transformer's self-attention mechanism to capture scene text information on both small and large patches.

However, it is not straightforward to apply the same Transformer-based encoder on the input patches with different sizes. Changing the patch resolution requires training a completely new encoder, thus resulting in parameter increases. To overcome this limitation, we propose to use a novel patch embedding strategy for normalizing the input patches of different sizes without losing information. The normalized patch embeddings are input to the encoder with corresponding scale embeddings for multi-scale scene text image representation.

The combination of the multi-scale encoders, PD, and MVM creates a powerful scene text recognizer. Extensive experiments illustrate the effectiveness of the proposed ProphetSTR model and also show its superiority when compared with the state-of-the-art (SOTA) method.

In summary, the main contributions of this paper are fourfold:

- We propose a strong Transformer-based STR model, i.e., ProphetSTR, exploring the improvements on both visual representations and LMs.
- Multi-prediction streams are adopted in the prophet decoder, which could provide the credibility of the results by prediction consistency. Additionally, the unreliable predictions are refined by a novel proposed multi-modal verification module to boost the recognition accuracy using both visual and semantic features with a masking strategy.
- The encoder adopts a scalable patch embedding strategy for processing input with varying patch resolutions. It allows multi-scale feature extraction in scene text images without introducing extra parameters.

- The proposed model achieves SOTA performance on many STR benchmarks for both synthetic and real training data. Extensive experiments illustrate the effectiveness of different modules in ProphetSTR.

## 2 RELATED WORKS

### 2.1 Language-free STR Methods

Language-free methods usually focus on leveraging visual features without considering the relationship between characters, such as [7, 8, 29, 38]. CRNN [29] employed CNN and RNN to model sequential visual features, and then directly fed them into a CTC decoder for prediction. DAN [38] built a convolutional alignment module, which performed alignment operations from a visual perspective and avoided using historical decoding information, thereby eliminating misalignments caused by decoding errors. ViTSTR [8] mainly focused on ViT-alike visual model construction and adopted a ViT training scheme for STR. SVTR [7] was designed to perceive inter-character and intra-character patterns using a mixture of global and local blocks, resulting in multi-grained character component perception within a single visual model.

Some methods were implemented through segmentation. The segmentation-based STR methods utilized Fully Convolutional Networks (FCN) to perform pixel-level character segmentation. Liao et al. [17] employed a pixel grouping approach to recognize characters by forming text regions from the segmented pixels. Textscanner [34] introduced an additional segmentation map that accurately transcribes characters in the correct order. Language-free methods struggle to effectively address recognition challenges in low-quality images due to their limited access to linguistic information.

### 2.2 Language-involved STR Methods

To incorporate linguistic information, many language-involved methods have been proposed. Most of those methods [18, 28] followed the one-way semantic transmission manner that guided the encoded visual features to attend the corresponding region with the help of semantic information of previous prediction. Some STR methods [8, 24, 44] used pre-trained LM or visually independent LM to correct the inaccurate recognition results with the text context in the image. SRN [44] proposed a global semantic reasoning module to consider global semantic context information and effectively combine it with visual context information to enhance prediction accuracy. ABINet [8] proposed autonomous, bidirectional, and iterative principles to guide the design of LM in STR.

Recently, many methods combined both visual and semantic modalities [1, 20, 39, 46] for STR. VisionLAN [39] proposed a language-aware mask to enhance the semantic features of visual information and proposed weakly-supervised complementary learning to generate accurate character-wise mask maps in MLM with only word-level annotations. JVSR [1] introduced a multi-stage character decoding paradigm with incremental refinement, where each stage utilized visual features for initial predictions, followed by a refinement step using combined visual-semantic information. MATRN [20] explored the combination of visual and semantic features extracted by visual model and LM and proposed an interactive component for multi-modal features enhancement with

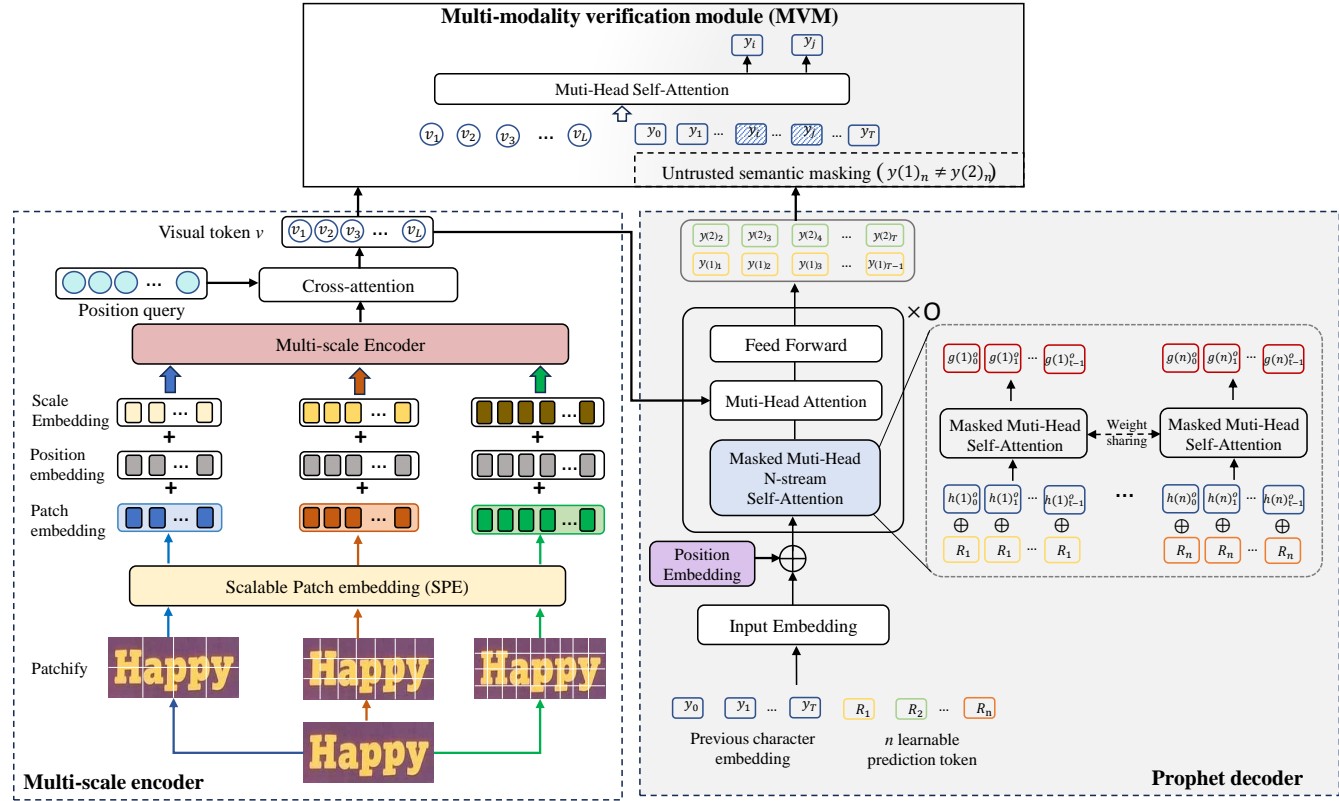

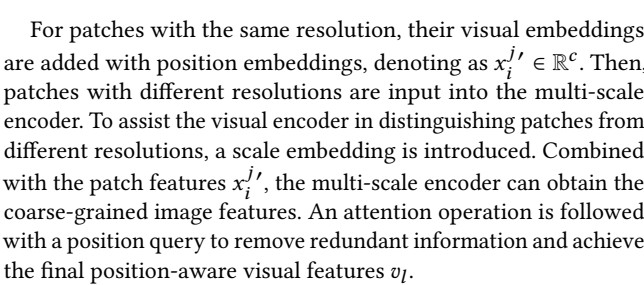

**Figure 1: The framework of our proposed model. The encoder adopts a scalable patch embedding strategy to normalize the patch embedding for different cropped sizes and integrates multi-scale patches to obtain powerful and fined visual representations. PD is responsible for character sequence prediction which could output multiple consecutive future predictions in parallel. MVM takes a further verification step to predict reliable results based on both visual and trusted semantic clues.**

bi-directional fusions. LPV [46] introduced a global linguistic reconstruction method to perceive the linguistic information in the visual space, which converted visual features into semantically rich ones gradually during the cascade process. Xiao et al. [41] related the language model with text length and proposed a dual character counting-aware visual and semantic modeling network for STR.

## 3 PROPOSED METHOD

### 3.1 Overall Architecture

The overall structure of the proposed method is present in Fig 2. It consists of three modules, multi-scale encoder, PD, and MVM.

Given an input scene text image $I \in \mathbb{R}^{W \times H \times C}$ ($W$, $H$ and $C$ represent the image width, height, and channels respectively), we first crop it to patches with different sizes. They are denoted as $P_i^j \in \mathbb{R}^{w_i \times h_i \times C}$, where $i \in \{1, \ldots, s\}$ and $j \in \{1, \ldots, n_i\}$. $s$ is the total patch scale. $n_i$, $w_i$ and $h_i$ is the patch numbers, width and height of scale $i$, respectively. A Scalable Patch Embedding (SPE) strategy is applied on those patches $P_i^j$ to tokenize them into patch embeddings $x_i^j \in \mathbb{R}^c$ based on the size $s$, where $c$ is the embedding dimension. The dimensions of the different patch embeddings are the same.

For patches with the same resolution, their visual embeddings are added with position embeddings, denoting as $x_i^{j\prime} \in \mathbb{R}^c$. Then, patches with different resolutions are input into the multi-scale encoder. To assist the visual encoder in distinguishing patches from different resolutions, a scale embedding is introduced. Combined with the patch features $x_i^{j\prime}$, the multi-scale encoder can obtain the coarse-grained image features. An attention operation is followed with a position query to remove redundant information and achieve the final position-aware visual features $v_l$.

Afterward, PD is used to predict the next $n$ character in parallel with the n-stream self-attention mechanism, which takes the embedding of the previously predicted character $y_{t-1}$ as query and the visual features $v_l$ as keys and values. $n$ special tokens are added with the input character embedding to indicate the steps of the next prediction.

The prediction manner of the PD may generate inconsistent results. They should not be trusted and required to be verified again. So, we mask those untrusted character embeddings to simulate the case of missing character-wise linguistic semantics. Finally, MVM takes both the masked prediction character embeddings and the position-aware visual features as input to predict the final character sequence in parallel.

## 3.2 Multi-scale Encoder

To better capture the character information with various text lengths in the image, we propose to model the input image with a multi-scale representation. Changing the patch resolution requires training new encoders, thus resulting in parameters increasing. To overcome this limitation, we use the SPE strategy for normalizing the input patches of different sizes without losing information.

The SPE process is displayed in Alg. 1. The patch embedding for the patch $P_i$ at scale $i$ is denoted as $x_i \in \mathbb{R}^c$. The superscript label $j$ of $x_i$ is ignored for a general patch embedding representation. We expect to find a "resized" patch embedding weights $\hat{w}$, so the patch representations before and after resizing remain the same, i.e., $\langle x_i, w \rangle = \langle x_i, \hat{w} \rangle$. After optimization, we find that the matrix to transform the patch embedding weights $w$ for a certain scale $i$ corresponds to the inverse of the bilinear resize operation $B$. The normalized patch embedding at any scale $i^*$ could be acquired by step 5 of Alg. 1. The detailed inference steps are present in the Appendix file.

The $j_{th}$ patch embeddings $x_i^j$ of patches $P_i^j$ at scale $i$ summing with its position embedding [6] $p_i^j$ and scale embeddings $S_i$ are input to the encoder $E_{ms}$, which stacks $l_1$ identical Transformer layers. It outputs the image tokens $\widetilde{x}_i$ capturing pixel, spatial and scale-level information:

$$\widetilde{x}_i = E_{ms}(x_1 + p_1 + S_1, \ldots, x_s + p_s + S_s). \tag{1}$$

Different scene text images might have their corresponding best resolutions for patch representation. Utilizing all the aggregated information of $\widetilde{x}_i$ is redundant, which hinders model learning and generalization. Therefore, we use length-fixed position queries [45] to extract positional-aware visual features through $l_2$ cross-attention layers:

$$p_{1:L}^{m+1} = \text{MHA}(p_{1:L}^m, \widetilde{x}_{1:s}^m, \widetilde{x}_{1:s}^m), \tag{2}$$

$p_l^0 \in \mathbb{R}^d$ ($l \in \{1, \ldots, L\}$) denote positional vectors, projecting from one-hot vectors at each location ranging from 0 to $L$. They are passed into the cross-attention module as the position query Q. The features of $\widetilde{x}_{1:s}$ are projected to the key K and value V. MHA(Q, K, V) denotes the multi-head attention operation [6]. $p_{1:L}^{m+1}$ is the output query after $m_{th}$ cross-attention layer. Finally, we get the positional aware visual tokens $v_{1:L}$ (namely $p_l^{l_2}$).

---

**Algorithm 1** Scalable Patch Embedding

**Input:** Patch embedding weights $w_i$ of patches $P_i$ at any pre-defined scale $i$. Patches $P_{i^*}$ at arbitrary scale $i^* \in \{i_1, \ldots, i_s\}$.
/*Define: patch embedding $x$, flatten operation $F$, linear mapping $B$*/
1: $x_i = F(P_i)^T F(w_i)$
2: **for** $i^* \leftarrow i_1$ to $i_s$ **do**
3:    **if** $i^* \neq i$ **then**
4:       $F(P_{i^*}) = B_i^{i^*} F(P_i)$     /*$B_i^{i^*}$ is the mapping from scale $i$ to $i^*$*/
5:       $x_{i^*} = F(P_{i^*})(B_i^{i^* T})^+ F(w_i)$    /*$(B)^+$ is the pseudo-inverse of matrix $B$*/
6:    **end if**
7: **end for**
   **Output:** Multi-scale patch embedding $x_{i^*}$, $i^* \in \{i_1, \ldots, i_s\}$

---

## 3.3 Prophet Decoder

For many Transformer-based decoders used in STR approaches [18, 32], their target is to estimate the probability of the next character given previous predictions and visual context $v_{1:L}$, namely $P(y_t|y_{<t}, v_{1:L})$. However, we expect the decoder to learn to predict future n-grams simultaneously, i.e., $P(y_{t:t+n-1}|y_{<t}, v_{1:L})$. In this way, the decoder can utilize previous contexts and also consider future clues for predicting the current results. Inspired by the ProphetNet [43] for word prediction, we propose a character PD adopting n-stream weight sharing self-attention block to achieve this goal.

As shown in Fig. 2, PD changes the normal masked multi-head self-attention block to a masked multi-head N-stream attention block. This block is based on the generally masked self-attention but is divided into $n$ parallel masked self-attention modules to predict the next $n$ consecutive future tokens respectively at each time step. Specifically, the $i_{th}$ prediction stream is responsible for modeling the probability $p(y_{t+i-1}|y_{<t}, v_{1:L})$. The input to PD is the embeddings of previously predicted character $y_{t-1}$ and $n$ learnable tokens $R_i$ ($i \in \{1, \ldots, n\}$) to indicate the order of prediction streams. Note that the N-stream attention block of ProphetNet is different from our PD. We find the mainstream attention of ProphetNet, which is the same as the masked multi-head self-attention in the traditional Transformer decoder for calculating the hidden states, is redundant and increases additional computational cost. Therefore, we remove it and use $n$ weight-sharing masked multi-head attention blocks as follows:

$$g(i)^o = \text{MHA}(h(i)^o, h(i)^o, h(i)^o), \tag{3}$$

where $h(i)^o = (h(i)_0^o, \cdots, h(i)_T^o)$, denoting the input sequence of the $i_{th}$ predicting stream in $o$-th layer of PD. $T$ is set as the maximum character sequence length. As the input of the first layer in Multi-Head N-stream Self-Attention, $h(i)_t^0$ is the concatenation of learnable stream indication token $R_i$ and the corresponding previous input character embeddings $y_{t-1}$ at time step $t$. During training, a lower triangular matrix is set to control each position to only focus on its previous tokens [8] for each prediction stream.

$g(i)^o$ is the output of the layer $o$ in $i_{th}$ masked self-attention-based block. It interacts with the visual tokens $v_{1:L}$ by cross-attention operation and the Feed Forward Network (FFN) to get the input $h(i)^{o+1}$ of next $(o + 1)_{th}$ layer of PD as:

$$\begin{aligned} g(i)^{o\prime} &= \text{MHA}(g(i)^o, v_{1:L}, v_{1:L}), \\ h(i)^{o+1} &= \text{LN}(g(i)^{o\prime} + \text{FFN}(g(i)^{o\prime})). \end{aligned} \tag{4}$$

The output of the last layer $O_{th}$, i.e., $h(i)^O$, is linear projected to predict the next $(i-1)_{th}$ output character $y_{t+i-1}$. Although the calculations of each stream are very similar, they are distinguished by different prediction steam tokens. Since each stream has the same structure, the weights can be shared among them.

Character in the same position will be predicted $n$ times by different streams. In each prediction stream, its referring context is different because the $i_{th}$ stream prediction sees previous context and future $n-i$ character clues. Therefore, the decoder may generate inconsistent results for predicting the same character in different streams. If so, they are considered unreliable predictions. Otherwise, they can be trusted.

## 3.4 Multi-modality Verification Module

Some STR methods [8, 47] refine the predicted character sequences using a combination of context-free vision and context-aware LMs. However, they are prone to erroneous rectification of correct initial predictions since the LM works conditional independently on the image features.

Our proposed PD provides a great criterion to determine the confidence of prediction results through the n-stream prediction way, which could avoid rectifying the initial correct predictions. Only the predicted unreliable results are required to be verified. Therefore, a further verification module, i.e., MVM, is proposed to re-predict the unreliable characters in parallel by referring to multi-modality vision and semantic information. It considers both the trusted characters with corresponding orders and position-aware visual features $v_{1:L}$ simultaneously, resulting in a more efficient and robust STR refinement process.

MVM has $l_3$ stacked Transformer layers, each of which consists of a self-attention layer and a feed-forward layer. Inspired by MLM in NLP models, the untrusted character predictions are replaced with special [mask] tokens, while the trusted ones are represented by their corresponding character embeddings. To better align with the position-aware visual features output from the encoders, the masked semantic embeddings $y_{1:T}^M$ are also added with the positional embeddings $p_{1:T}$ to indict the sequence order:

$$y_i^{M\prime} = y_i^M + p_i. \tag{5}$$

The position-aware visual tokens $v_{1:L}$ as well as the masked position-aware semantic features $y_{1:T}^M{}'$ are input to the MVM. Between them, a special separator token [SEP] is inserted to distinguish the two modalities. The two kinds of features are interacted through the self-attention mechanism as follows:

$$[v_{1:L}^{u+1}, y_{1:T}^{u+1}] = \text{MHA}([v_{1:L}^u, y_{1:T}^u], [v_{1:L}^u, y_{1:T}^u], [v_{1:L}^u, y_{1:T}^u]), \tag{6}$$

The superscript $u \in \{1, \dots, l_3\}$) denotes the $u_{th}$ self-attention layer. In the first layer, $y_{1:T}^0 = y_{1:T}^M$ and $v_{1:L}^0 = v_{1:L}$. In the last Transformer layer, we only use the interacted semantic features $y_{1:T}^{l_3}$ for character prediction.

This step is implemented by a parallel linear transformation. Concretely, a fully connected layer and softmax operation are employed to generate a transcript sequence of size $T$, where ideally, the prediction results of trusted characters remain the same, while each of the masked characters is classified into a certain character category. The output character sequence is determined by incorporating both trusted multi-modality features, thus being more reliable than that of PD. It is noted that the masked semantics are reasoned in a parallel way for a single time in the MVM, making it run efficiently to generate more accurate results.

## 3.5 Loss Function

The loss function is composed of two items. Since the prediction step by PD has $n$ different streams to predict the future $n$ characters, each prediction stream adopts an independent cross-entropy loss

for training. This objective function is formalized as:

$$
\begin{aligned}
\mathcal{L}_{PD} &= -\sum_{j=0}^{n-1} \lambda_j \left( \sum_{t=1}^{T-j} \log p_\theta(y_{t+j}|y_{<t}, v) \right) \\
&= -\lambda_0 \left( \sum_{t=1}^{T} \log p_\theta(y_t|y_{<t}, v) \right) \\
&\quad - \sum_{j=1}^{n-1} \lambda_j \left( \sum_{t=1}^{T-j} \log p_\theta(y_{t+j}|y_{<t}, v) \right),
\end{aligned}
\tag{7}
$$

where $\theta$ is the trainable parameters of the model. The first item of the second equation in Eq.(7) is a traditional LM to learn the next prediction, and the second item is the objective for future predictions. $\lambda_j$ is the weight of $j_{th}$ stream to balance the importance of different prediction streams.

The cross-entropy loss is also used to supervise the learning of MVM, which is formulated by:

$$\mathcal{L}_{MVM} = -\frac{1}{T} \sum_{t=1}^{T} \log p_\theta(y_t|y_{1:T}^M, v). \tag{8}$$

## 4 EXPERIMENTS

### 4.1 Datasets

MJSynth (MJ) [13] and SynthText (ST) [10] are synthetic text image datasets, which contain 9 million and 8 million text images respectively. We use their union as our training data. Six common benchmarks contain three regular text datasets and three irregular text datasets are used for the test. The regular text datasets are IIIT5K-Words (IIIT5K) [19], Street View Text (SVT) [37] and ICDAR 2013 (IC13) [16]. The irregular text datasets are ICDAR 2015 (IC15) [15], Street View Text Perspective (SVTP) [22], and CUTE 80 (CUTE) [27]. Images in these datasets are most curved and distorted. WordArt dataset (artistic text) [42] is used for testing to evaluate the generalization of scene text recognizer.

In addition, we also train and test the model with the real dataset Union14M-L [26], which contains 3.2 million samples collected from publicly available datasets. The test set of Union-14M-Benchmark contains 0.4M samples in a variety of situations, such as Curve, Multi-Oriented, Artistic, Contextless, Salient, Multi-Words, and General.

### 4.2 Implementation Details

We first resize the text images to $36 \times 144$ and use typical image processing ways, e.g., adding Gaussian noise, rotation, perspective distortion, and motion blur, for data augmentation, To train our model, the initial learning rate is set to $4 \times 10^{-4}$. The first 10k iterations are used for warm-up. The whole training iterations are determined as: $lr = d_{model}^{-0.5} \cdot min(n^{-0.5}, n \cdot warm_n^{-1.5})$, where $n$ and $warm_n$ represent the number of normal iterations and warm iterations. $d_{model}$ is set to 384. The maximum length of the model output sequence $T$ is set to 26. Experimentally, we set $l_1=l_2=l_3=4$ and $O=4$, respectively. The hyper-parameters $\lambda_j$ are set equally. To train MVM, we mask 50% predicted characters of PD randomly. During inference, we use a lowercase alphanumeric charset, and the number of character classes is set to 36. The models were trained

**Table 1: Accuracy(%) and speed (ms/image) of models with different combinations of patch resolutions.**

| Patch size | III5K | SVT | IC13 | Avg. | IC15 | SVTP | CUTE | Avg. | Speed |
|---|---|---|---|---|---|---|---|---|---|
| $E_6$ | 95.5 | 94.9 | 96.2 | 95.53 | 85.6 | 90.8 | 90.3 | 88.90 | 21.52 |
| $E_9$ | 94.8 | 94.4 | 96.2 | 95.13 | 85.3 | 89.1 | 88.1 | 87.50 | 14.21 |
| $E_{12}$ | 94.0 | 94.6 | 96.4 | 95.00 | 84.4 | 90.4 | 86.1 | 86.97 | 8.00 |
| $E_{18}$ | 91.8 | 92.1 | 95.8 | 93.23 | 83.7 | 87.1 | 85.3 | 85.37 | 7.84 |
| $E_{9,12}$ | 95.6 | 94.9 | 96.9 | 95.80 | 86.0 | 90.8 | 90.5 | 89.10 | 20.16 |
| $E_{9,12,18}$ | 96.6 | 95.1 | 97.2 | 96.30 | 86.2 | 91.2 | 92.7 | 90.03 | 23.40 |
| $E_{6,9,12,18}$ | 97.4 | 95.1 | 97.4 | 96.63 | 87.0 | 92.4 | 94.1 | 91.17 | 29.57 |

on 2 NVIDIA A40, with a batch size of 1280. The training epochs are 8 for synthetic data and 20 for real data.

### 4.3 Ablation Study

*4.3.1 Patch Size.* This section presents a controlled patch size study of our proposed model by evaluating the STR performance on the aspects of effectiveness and efficiency. We input single-size patches and a multi-scale combination of them for training and testing. For a fair comparison, we set the prediction stream number to 2 in all those experiments and only observed the results of the first stream in PD. Four patch sizes, i.e., 6×6, 9×9, 12×12, 18×18 pixels are used for this ablation study. We denote the model as $E_i$ according to the input patch size, where $i \in \{6, 9, 12, 18\}$.

Results are shown in Table. 1. The first four rows show the results using different single-scale patch embedding in the encoders. We can see that the smaller the patch resolution, the higher the accuracy rate. The model $E_6$ achieves the highest average accuracy rate on both regular datasets and irregular datasets than other single-scale models, but its inference time is the slowest. Model $E_9$ gets comparable performance with model $E_6$. But, it takes half as much time for inference. The averaged recognition accuracy of $E_{12}$ and $E_9$ are approximately equal. With the patch resolution becoming larger, the recognition accuracy model drops gradually.

The last three rows in Table. 1 presents the results that take multi-scale patch sizes as input. We can see that the model $E_{9,12}$ surpasses $E_6$ on almost all the datasets with less computational time. With more patch resolutions used in the encoders, the recognition accuracy is increasing continually. This demonstrates that integrating multi-resolution patch tokens is important for effectively capturing scene text image information.

*4.3.2 Number of Prediction Streams.* To explore the effectiveness of predicting n-gram, we compare our model with setting n=1, 2, and 3 in the PD. Two encoders, $E_{12}$ and $E_{9,12,18}$, are used to remove the influence brought by the proposed image encoding module.

**Table 2: Accuracy(%) of models with different decoders.**

| Model | III5K | SVT | IC13 | IC15 | SVTP | CUTE | Avg. |
|---|---|---|---|---|---|---|---|
| $E_{12}+PD_1$ | 94.0 | 92.0 | 96.4 | 83.0 | 88.1 | 85.3 | 89.80 |
| $E_{12}+PD_2$ | 94.0 | 94.6 | 96.4 | 84.4 | 90.4 | 86.1 | 90.99 |
| $E_{12}+PD_3$ | 94.4 | 94.9 | 96.1 | 84.6 | 88.9 | 87.2 | 91.02 |
| $E_{9,12,18}+PD_1$ | 94.2 | 94.6 | 96.3 | 84.8 | 88.4 | 89.1 | 91.23 |
| $E_{9,12,18}+PD_2$ | 96.6 | 95.1 | 97.2 | 86.2 | 91.2 | 92.7 | 93.16 |
| $E_{9,12,18}+PD_3$ | 96.7 | 95.1 | 97.8 | 85.8 | 90.9 | 92.2 | 93.08 |

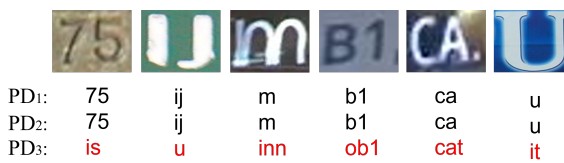

| | | | | | | |
|---|---|---|---|---|---|---|
| $PD_1$: | 75 | ij | m | b1 | ca | u |
| $PD_2$: | 75 | ij | m | b1 | ca | u |
| $PD_3$: | is | u | inn | ob1 | cat | it |

**Figure 2: The recognition results of models with different PD. The incorrectly recognized character is marked red.**

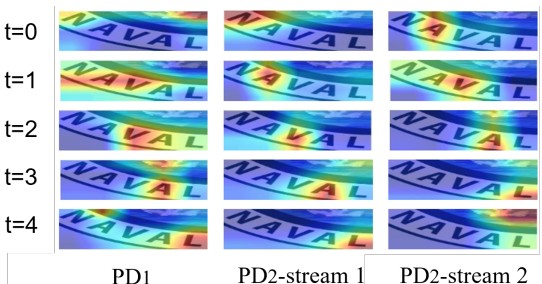

**Figure 3: The character attention in the decoders of $E_{12}+PD_1$ and $E_{12}+PD_2$ at each time step.**

They are connected with different prophet decoders $PD_i$, where $i$ indicates the number of prediction streams in the proposed decoder. Results are shown in Table. 2.

We can see that the performance of $PD_2$ and $PD_3$ is better than $PD_1$ on all of the six datasets whatever the encoder is. It demonstrates that using a n-gram (n>1) character prediction strategy is better than a single-gram one. However, the performance of using higher grams for character prediction in scene text recognition tasks might be decreased. The gap between the 2-gram model and the 3-gram model is not large. We check the incorrectly recognized text and find that the 3-gram model has the problem of recognizing short text. We list some examples in Fig. 2. In addition, the computational and time cost becomes higher with the increasing prediction streams in PD. The inference time of $E_{9,12,18}$ plus three different prediction streams (i.e, $PD_1$, $PD_2$ and $PD_3$ ) are 20.95ms, 23.40ms and 26.22ms, respectively. Comprehensively, we select $PD_2$ as our decoder in other experiments.

Some visualization of the character's attention in different prediction streams of the decoder is present in Fig. 3. The $i_{th}$ row displays the attention at time step $i$-1. We can see that the model of $E_{12}+PD_2$ could pay attention to the areas of corresponding characters more correctly. The second stream of $PD_2$ could also pay high attention correctly to the next character during prediction. It could promote the concentration of character attention for the first stream.

*4.3.3 Effect of Different Modules.* To evaluate the effect of different modules of our proposed method, we conduct experiments on eight separate models and display the results in Table. 5. If a multi-scale encoder (i.e., $E_{ms}$) is not selected, it represents we only use a single-scale encoder $E_{12}$ as input. Otherwise, we adopt $E_{9,12,18}$. If PD is not selected, it means we only use $PD_1$ as a decoder which degrades to a normal Transformer-based decoder. If it is selected, that means the model uses the two-stream $PD_2$. If MVM is not selected, the

**Table 3: Accuracy comparison with SOTA STR methods on six common benchmarks and WordArt datasets. Models are trained on synthetic datasets (MJ and ST)**

| Methods | Training Data | Regular | | | | Irregular | | | | Artistic |
|---|---|---|---|---|---|---|---|---|---|---|
| | | IIIT5k | SVT | IC13 | Avg | IC15 | SVTP | CUTE | Avg | WordArt |
| ASTER(2019) [30] | ST+MJ | 93.4 | 89.5 | 91.8 | 91.6 | 76.1 | 78.5 | 79.5 | 78.0 | 57.9 |
| NRTR(2019) [28] | ST+MJ | 90.1 | 91.5 | 95.8 | 92.5 | 79.4 | 86.6 | 80.9 | 82.3 | 58.5 |
| RobustScanner(2020) [45] | ST+MJ | 95.3 | 88.1 | 94.8 | 92.7 | 77.1 | 79.5 | 90.3 | 82.3 | 61.3 |
| SEED(2020) [24] | ST+MJ | 93.8 | 89.6 | 92.8 | 92.1 | 80.0 | 81.4 | 83.6 | 81.7 | 60.1 |
| SRN(2020) [44] | ST+MJ | 94.8 | 91.5 | 95.5 | 93.9 | 82.7 | 85.1 | 87.8 | 85.2 | - |
| VisionLAN(2021) [39] | ST+MJ | 95.8 | 91.7 | 95.7 | 94.4 | 83.7 | 86.0 | 88.5 | 86.1 | - |
| ABINet-LV(2021) [8] | ST+MJ+WiKi | 96.2 | 93.5 | 97.4 | 95.7 | 86.0 | 89.3 | 89.2 | 88.2 | 67.4 |
| S-GTR(2021) [11] | ST+MJ | 95.8 | 94.1 | 96.8 | 95.6 | 84.6 | 87.9 | 92.3 | 88.3 | - |
| PTIE(2022) [32] | ST+MJ | 96.3 | 94.9 | 97.2 | 96.1 | 84.3 | 90.1 | 91.7 | 88.7 | - |
| MATRN(2022) [20] | ST+MJ | 96.6 | 95.0 | 95.8 | 95.8 | 82.8 | 90.6 | 93.5 | 89.0 | - |
| CornerTransformer(2022) [42] | ST+MJ | 95.9 | 94.6 | 96.4 | 95.6 | 86.3 | 91.5 | 92.0 | 89.9 | 70.8 |
| CDistNet(2023) [48] | ST+MJ | 96.4 | 93.5 | 97.4 | 95.8 | 86.0 | 88.7 | 93.4 | 89.4 | - |
| LISTER(2023) [3] | ST+MJ | 96.9 | 93.8 | 97.9 | 96.2 | 87.5 | 89.6 | 90.6 | 89.2 | - |
| LPV(2023) [46] | ST+MJ | 97.3 | 94.6 | 97.3 | 96.4 | 87.5 | 90.9 | 94.8 | 91.0 | - |
| Count-aware STR(2024) [41] | ST+MJ | 96.6 | 93.6 | 97.2 | 95.8 | 84.8 | 89.0 | 92.4 | 88.7 | 68.2 |
| ProphetSTR$_{9,12,18}$(E$_{9,12,18}$+PD$_2$+MVM) | ST+MJ | 96.6 | 95.1 | 97.2 | 96.3 | 86.2 | 91.2 | 92.7 | 90.0 | 72.5 |
| ProphetSTR$_{6,9,12,18}$(E$_{6,9,12,18}$+PD$_2$+MVM) | ST+MJ | 97.4 | 95.1 | 97.4 | 96.6 | 87.0 | 92.4 | 94.1 | 91.2 | 72.8 |

**Table 4: Accuracy comparison with SOTA STR methods on six common benchmarks and Union14M-benchmark. The models are trained on the training set of Union14M-L.**

| Methods | Common Benchmarks | | | | | | | Union14M-benchmark | | | | | | | |
|---|---|---|---|---|---|---|---|---|---|---|---|---|---|---|---|
| | IIIT5k | SVT | IC13 | IC15 | SVTP | CUTE | Avg | Curve | Multi-Oriented | Artistic | Contextless | Salient | Multi-Words | General | Avg |
| CRNN [29] | 90.8 | 83.8 | 91.8 | 71.8 | 70.4 | 80.9 | 81.6 | 19.4 | 4.5 | 34.2 | 44.0 | 16.7 | 35.7 | 60.4 | 30.7 |
| ASTER [30] | 94.3 | 88.9 | 92.6 | 77.7 | 80.5 | 86.5 | 86.7 | 38.4 | 13.0 | 41.8 | 52.9 | 31.9 | 49.8 | 66.7 | 42.1 |
| NRTR [28] | 96.2 | 94.0 | 96.9 | 80.9 | 84.8 | 92.0 | 90.8 | 49.3 | 40.6 | 54.3 | 69.6 | 42.9 | 75.5 | 75.2 | 58.2 |
| SAR [12] | 96.6 | 92.4 | 96.0 | 82.0 | 85.7 | 92.7 | 90.9 | 68.9 | 56.9 | 60.6 | 73.3 | 60.1 | 74.6 | 76.0 | 67.2 |
| SATRN [14] | 97.0 | 95.2 | 97.9 | 87.1 | 91.0 | 96.2 | 93.9 | 74.8 | 64.7 | 67.1 | 76.1 | 72.2 | 74.1 | 75.8 | 72.1 |
| RobustScanner [45] | 96.8 | 92.4 | 95.7 | 86.4 | 83.9 | 93.8 | 91.2 | 66.2 | 54.2 | 61.4 | 72.7 | 60.1 | 74.2 | 75.7 | 66.4 |
| SRN [44] | 95.5 | 89.5 | 94.7 | 79.1 | 83.9 | 91.3 | 89.0 | 49.7 | 20.0 | 50.7 | 61.0 | 43.9 | 51.5 | 62.7 | 48.5 |
| VisionLAN [39] | 96.3 | 91.3 | 95.1 | 83.6 | 85.4 | 92.4 | 91.3 | 70.7 | 57.2 | 56.7 | 63.8 | 67.6 | 47.3 | 74.2 | 62.5 |
| ABINet [8] | 97.2 | 95.7 | 97.2 | 87.6 | 92.1 | 94.4 | 94.0 | 75.0 | 61.5 | 65.3 | 71.1 | 72.9 | 59.1 | 79.4 | 69.2 |
| SVTR [7] | 95.9 | 92.4 | 95.5 | 83.9 | 85.7 | 93.1 | 91.1 | 72.4 | 68.2 | 54.1 | 68.0 | 71.4 | 67.7 | 77.0 | 68.4 |
| MATRN [20] | 98.2 | 96.9 | 97.9 | 88.2 | 94.1 | 97.9 | 95.5 | 80.5 | 64.7 | 71.1 | 74.8 | 79.4 | 67.6 | 77.9 | 74.6 |
| MAERec-S [26] | 97.4 | 95.7 | 97.3 | 86.7 | 91.0 | 96.2 | 94.1 | 75.4 | 66.5 | 66.0 | 76.1 | 72.6 | 77.0 | 80.8 | 73.5 |
| ProphetSTR$_{9,12,18}$ | 97.6 | 97.2 | 97.8 | 89.4 | 94.1 | 96.8 | 95.5 | 80.5 | 64.2 | 72.4 | 74.3 | 77.5 | 64.8 | 80.2 | 73.4 |
| ProphetSTR$_{6,9,12,18}$ | 98.4 | 98.0 | 97.8 | 90.2 | 94.1 | 97.9 | 96.1 | 82.4 | 66.5 | 72.4 | 75.5 | 77.5 | 67.9 | 82.5 | 75.0 |

**Table 5: Effect of different modules for STR.**

| MS | PD | MVM | III5K | SVT | IC13 | IC15 | SVTP | CUTE | Avg. |
|---|---|---|---|---|---|---|---|---|---|
| | | | 93.2 | 90.4 | 91.8 | 79.6 | 83.4 | 85.4 | 87.30 |
| ✓ | | | 94.0 | 93.5 | 95.7 | 84.3 | 87.6 | 87.2 | 90.38 |
| | ✓ | | 93.9 | 94.3 | 95.3 | 82.7 | 86.1 | 88.2 | 89.86 |
| | | ✓ | 94.0 | 92.0 | 96.4 | 83.0 | 88.1 | 85.3 | 89.80 |
| ✓ | ✓ | | 94.9 | 94.4 | 95.9 | 85.0 | 90.1 | 89.2 | 91.58 |
| ✓ | | ✓ | 94.2 | 94.6 | 96.3 | 84.8 | 88.4 | 89.1 | 91.23 |
| | ✓ | ✓ | 94.0 | 94.6 | 96.4 | 84.4 | 90.4 | 86.1 | 90.99 |
| ✓ | ✓ | ✓ | 96.6 | 95.1 | 97.2 | 86.2 | 91.2 | 92.7 | 93.16 |

model is composed of encoder E$_{9,12,18}$ and decoder PD$_2$. We get the results of the first stream in PD$_2$ for comparison.

The results in the first row are the baseline which is a simple encoder-decoder-based STR model. The following three rows present the performance of individual effects of multi-scale visual encoder, PD, and MVM. We can see that they improve the averaged accuracy by 2.92%, 2.56%, and 2.5% respectively compared with the baseline model, demonstrating their positive effect for boosting the recognition performance separately.

The results from $5_{th}$ to $7_{th}$ rows display the pair-wise combination effect of different modules respectively. The recognition rates are increment. Among them, the model adopting a multi-scale strategy and PD gets the highest accuracy. It illustrates our proposed encoder and decoder model could generate relatively reliable results. After combining the three modules, the ensemble model brings about a further performance boost. The results of the ensemble model are better than that of any pair-wise module combined models, which also proves the importance of each module for STR. Additionally, the average inference time of the ensemble model is 32.82ms. Among them, the modules E$_{6,9,12,18}$+PD$_2$ take about 29.57ms and MVM takes about 3ms.

## 4.4 Comparison with State-of-the-Art Methods

We first compare our proposed full models with the state-of-the-art (SOTA) methods on seven STR benchmark datasets, including the artistic text dataset WordArt. ProphetSTR$_{9,12,18}$ represents the full

**Table 6: Accuracy(%) of output from different streams in PD and MVM. The Model is trained on the training set of Union14M-L.**

| Output | Common Benchmarks | | | | | | | Union14M-benchmark | | | | | | | |
|---|---|---|---|---|---|---|---|---|---|---|---|---|---|---|---|
| | IIIT5k | SVT | IC13 | IC15 | SVTP | CUTE | Avg | WordArt | Curve | Multi-Oriented | Artistic | Contextless | Salient | Multi-Words | General | Avg |
| PD2-Stream1 | 97.2 | 96.9 | 96.9 | 86.9 | 93.4 | 94.4 | 94.3 | 73.5 | 75.0 | 63.2 | 71.1 | 73.3 | 74.6 | 63.5 | 77.8 | 71.2 |
| PD2-Stream2 | 87.9 | 83.4 | 88.4 | 73.2 | 76.6 | 78.6 | 81.3 | 59.6 | 49.5 | 40.6 | 54.3 | 62.3 | 63.3 | 51.5 | 72.5 | 56.3 |
| MVM | 98.4 | 98.0 | 97.8 | 90.2 | 94.1 | 97.9 | 96.1 | 75.3 | 82.4 | 66.5 | 72.4 | 75.5 | 77.5 | 67.9 | 82.5 | 75.0 |

| | | | |
|---|---|---|---|
| PD2-stream1: | autwmm | jeweless | harry |
| PD2-stream2: | utumn | eveless | appy |
| MVM: | autumn | jewelers | harry |

**Figure 4: The prediction results of the first and second stream of PD, and MVM.**

model of $E_{9,12,18}$+PD$_2$+MVM, and ProphetSTR$_{6,9,12,18}$ represents that of $E_{6,9,12,18}$+PD$_2$+MVM. All the comparing methods are trained on MJSynth and SynthText. Results can be found in Table 3. Our method achieves superior performance compared with the other methods. Notably, our proposed ProphetSTR$_{6,9,12,18}$ obtains the highest average scores on all the datasets.

When we compare the model ProphetSTR$_{6,9,12,18}$ with others trained on Union14M (without pre-training), our method could also achieve SOTA performance. It almost obtains the best performance on all the datasets of common benchmarks, and performs the best, especially for curve, artistic, and general scene text for the Union14M dataset.

Compared with language-free STR methods, e.g., CRNN [29], ASTER [30], and some language-aware STR methods, e.g., SRN [44], ABINet [8], VisonLAN [39] and LPV [46], our model has shown significant improvements on almost all the datasets. It implies the strong recognition capabilities of our proposed model. MATRN [20] combines visual and semantic features for AR STR. But, instead of masking semantics, it masks regions on the image. Compared with it, our method outperforms it on most datasets when trained with synthesis data and achieves comparable results when trained with real data. It further illustrates the effectiveness of using multi-modality information for STR.

## 4.5 Recognition Accuracy of Different Modules

In this section, we show the recognition accuracy of different streams in PD and MVM on different test datasets by training with the Union14M-L dataset. Note that the second stream of PD starts the prediction from the second character of the text images. Thus, we count the accuracy of this stream by ignoring the first character of ground truth. The results are shown in Table. 6. We can see that the recognition accuracy of the first stream is much higher than that of the second stream in PD. It may rise from the reason that the output from the second stream refers to the previous semantic information for predicting character at $t$+1 time step. In contrast, the first stream refers to both previous predictions and future $t$+1 information for predicting the output of time step $t$. The first stream could collect more clues than the second stream in PD. After adding the verification module, we can see that MVM can improve the recognition performance on all the datasets, especially

**Table 7: Character Recognition Inconsistency Rate of Prophet Decoder**

| Training Data | III5K | SVT | IC13 | IC15 | SVTP | CUTE |
|---|---|---|---|---|---|---|
| MJ+ST | 16.5 | 16.2 | 11.7 | 26.9 | 16.2 | 21.8 |
| Union14M-L | 12.2 | 14.0 | 8.8 | 24.5 | 15.0 | 18.7 |

for the Curve data of the Union14M benchmark, which boosts the recognition rate by 7.4%. It demonstrates the essential effectiveness of the MVM for recognition verification.

Some prediction results of the PD and MVM are positioned in Fig. 4. We can see the inconsistent predictions of the two streams in the PD are corrected by MVM.

## 4.6 Character Recognition Inconsistency Rate of Prophet Decoder

Since the input of MVM is based on the inconsistency of the outputs from different streams in PD, we further count the character-level inconsistency recognition rate f. The *inconsistency rate* is computed by:

$$Inconsistency\ Rate = \frac{\sum \mathbb{1}(Pred_{stream2} \neq Pred_{stream1})}{\text{Number of all characters}}, \quad (9)$$

where the numerator is the accumulated numbers of inconsistent character recognition, results in the two streams of PD. Table. 7 shows the results of the models trained on different datasets. We find that this rate is larger on irregular datasets. Compared with models trained on synthetic data, training on the real dataset, i.e., Union14M, can reduce the recognition inconsistency and obtain more accurate results. However, this great *inconsistency rate* motivates us to propose MVM for boosting the performance and eliminating the unreliable character recognition results. More inconsistent recognition results of PD$_2$ and the MVM rectified results are displayed in the Appendix file.

## 5 CONCLUSION

In this paper, we proposed a novel scene text recognizer, Prophet-STR, which consisted of a multi-scale encoder, a prophet decoder (PD), and a multi-modality verification module (MVM). The encoders employed a scalable patch embedding strategy to handle inputs with different patch resolutions, allowing multi-scale patch information aggregated image representation. PD predicted the next consecutive steps of characters by utilizing future information. MVM could generate more accurate results based on both visual features and reliable semantic clues. By assembling the three modules, ProphetSTR shows powerful performance on different datasets.

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
