# OpenReview forum: "Trust Prophet or Not? Taking a Further Verification Step toward Accurate Scene Text Recognition"
_acmmm.org/ACMMM/2024/Conference — MM2024 Poster_

### Official Review · Reviewer_hLMm · 2024-05-24

**Rating:** 3
**Confidence:** 4

**Summary:**

The paper introduces a scene text recognition (STR) model named ProphetSTR. The model incorporates a multi-scale encoder, a prophet decoder (PD), and a multi-modality verification module (MVM) to enhance the accuracy of scene text recognition. The multi-scale encoder handles inputs with varying patch resolutions, the PD predicts characters using future information, and the MVM refines predictions based on visual features and reliable semantic clues. The paper reports state-of-the-art performance on multiple benchmarks and an extensive set of experiments to validate the contributions of different components.

**Strengths:**

1. The paper presents a unique approach to STR by introducing a prophet decoder that utilizes future information for character prediction.

2. The model achieves state-of-the-art results on multiple benchmarks, indicating a significant advancement in the field of scene text recognition.

**Limitations:**

1. The description of the Prophet Decoder is unclear. It is not well understood how the authors implement the masked multi-head N-stream attention block, and how the PD decodes the output sequence from the n streams during the inference phase. Please provide a detailed explanation.

2. The output of the Prophet Decoder in Figure 1 is depicted as $(y^{2}_2, y^{2}_3, y^{2}_4, \ldots, y^{2}_T)$ and $(y^{1}_1,y^{1}_2,y^{1}_3,\ldots,y^{1}_{T-1}$. According to the paper, there should be two predictions at each position, but at time step 1, only one prediction is shown. Could you please clarify this discrepancy?

**Suitability:**

3

---

### Official Review · Reviewer_Wmzp · 2024-05-26

**Rating:** 3
**Confidence:** 3

**Summary:**

This paper introduces ProphetSTR, a novel scene text recognition (STR) model that leverages linguistic knowledge and visual context for improved accuracy. Unlike traditional models, ProphetSTR uses an n-stream self-attention mechanism to predict characters based on both past and near-future context. It includes a multi-modality verification module to mask unreliable predictions and a multi-scale weight-sharing encoder for enhanced image representation. Extensive experiments show that ProphetSTR achieves state-of-the-art performance across various benchmarks, validating the effectiveness of its innovative components.

**Strengths:**

The idea of predicting n-gram with a verification step sounds interesting.

**Limitations:**

It would be beneficial to provide a clearer explanation of the motivation behind predicting n-grams with the verification step and how this approach offers advantages over 1-gram prediction.

Additionally, while the improvement compared to other state-of-the-art methods is evident, it appears relatively modest.

**Suitability:**

2

---

### Official Review · Reviewer_5MNA · 2024-06-07

**Rating:** 4
**Confidence:** 3

**Summary:**

The authors propose ProphetSTR, a novel Scene Text Recognition (STR) model that uses an n-stream attention mechanism to predict multiple characters simultaneously, leveraging both past and near future semantic clues for improved accuracy. They introduce a multi-modality verification module to handle unreliable predictions by combining visual and trusted semantic features. Additionally, a multi-scale weight-sharing encoder is used for detailed image representation. ProphetSTR demonstrates state-of-the-art performance on benchmarks, validated by extensive experiments and ablation studies.

**Strengths:**

1. Prophet Decoder for Future n-Gram Prediction: ProphetSTR's prophet decoder (PD) predicts future n-gram characters simultaneously. This n-step-ahead prediction guides the model to plan for future tokens while considering both previous context and future clues.
2. Multi-Modal Verification Module: a multi-modal verification module that uses vision features and trusted outputs to verify predictions.
3. Transformer-Based Multi-Scale Encoder: ProphetSTR uses multi-scale patch representations to capture scene text information at various scales by normalizing input patches of varying sizes without losing information and inputs them to the encoder with corresponding scale embeddings.

**Limitations:**

1. It is not clear what brings the difference between 1-gram and n-gram and the motivation for using n-gram from the paper. Moreover, the idea of "predicting the next 𝑛 consecutive future tokens respectively at each time step" can be seen in the graph transformer problem, where at the current time (Tc), the model can predict the next n time slices after (Tc+1, Tc+2, ... Tc+n). As the core of the proposed method, this point should have more discussion and explanation.
2. Is there any strategy for generating masks in the n-parallel masked multihead attention block?

**Suitability:**

2

---

### Meta-Review · Area_Chair_YXor · 2024-06-26

**Recommendation:** Accept (Poster)
**Confidence:** 4

**Metareview:**

The reviewers have come up to the following strengths and limitations

STRENGTH
- Prophet Decoder for Future n-Gram Prediction
- Multi-Modal Verification Module
- Transformer-Based Multi-Scale Encoder
- Innovative Approach to STR
- State-of-the-Art Results

LIMITATIONS
- Lack of Clarity on n-Gram Motivation (it is better after the rebuttal)
- Predicting Future Tokens needs more discussion
- Unclear Mask Generation Strategy (it is better after the rebuttal)
- Unclear Verification Step Motivation
- Modest Improvement Over SOTA
- Unclear Description of Prophet Decoder
- Discrepancy in Output Depiction

The rebuttal did well, and all reviewers agreed to raise their rating. Average rating 3.33